# Sea Buckthorn Flavonoid Extracted by High Hydrostatic Pressure Inhibited IgE-Stimulated Mast Cell Activation through the Mitogen-Activated Protein Kinase Signaling Pathway

**DOI:** 10.3390/foods13040560

**Published:** 2024-02-12

**Authors:** Zhuomin Yan, Xiaoping Feng, Xinian Li, Zhenpeng Gao, Zhouli Wang, Guangxu Ren, Fangyu Long

**Affiliations:** 1College of Food Science and Engineering, Northwest A&F University, Yangling 712100, China; yanzhuomin7@163.com (Z.Y.); xiaopingfeng0220@163.com (X.F.); 15773123313@163.com (X.L.); gzp5988@163.com (Z.G.); wzl1014@nwsuaf.edu.cn (Z.W.); 2Institute of Food and Nutrition Development, Ministry of Agriculture and Rural Affairs of the People’s Republic of China, Beijing 100081, China; renguangxu@caas.cn

**Keywords:** allergy, sea buckthorn flavonoid, HHP, RBL-2H3, MAPK

## Abstract

Sea buckthorn (*Hippophaë rhamnoides* L.), as one of the Elaeagnaceae family, has the significant function of anti-tumor, anti-inflammation, anti-oxidation, and other physiological activities. High hydrostatic pressure (HHP) extraction has the advantages of being easy and efficient, while maintaining biological activity. In this study, sea buckthorn flavonoid (SBF) was extracted with HHP and purified sea buckthorn flavonoid (PSBF) was isolated by AB-8 macroporous resin column. HPLC analysis was used to quantified them. In addition, the effect of anti-allergy in RBL-2H3 cells by SBF, PSBF, and their flavonoid compounds was evaluated. The results demonstrate the conditions for obtaining the maximum flavonoid amount of SBF: 415 MPa for 10 min, 72% ethanol concentration, and a liquid to solid ratio of 40 mL/g, which increased the purity from 1.46% to 13.26%. Both SBF and PSBF included rutin, quercitrin, quercetin, isorhamnetin, and kaempferol. In addition, quercitrin, kaempferol, and SBF could regulate Th1/Th2 cytokine balance. Moreover, extracellular Ca^2+^ influx was reduced by quercitrin and PSBF. Furthermore, rutin, quercetin, iso-rhamnetin, and SBF could also inhibit P-p38 and P-JNK expression, thereby suppressing the phosphorylation of the MAPK signaling pathways. Overall, SBF is effective for relieving food allergy and might be a promising anti-allergic therapeutic agent.

## 1. Introduction

Up to 20% of the population suffer from allergic disease, and the prevalence of food allergies is on the rise globally [1]. When the interaction of allergen-specific IgE with its high-affinity IgE receptor (FcεRI) happens, establishing receptor cross-linking on sensitized mast cells or basophils and turn on to release pro-inflammatory mediators or cytokines, food allergic reactions occurs [2]. Upon allergen provocation, mast cells engender many kinds of inflammatory mediators, inducing immune responses to various external stimuli in the body [3]. Cytokines play essential roles in IgE immunoglobulin conversion on basophils, such as IL-4, which can lead to proliferation and differentiation in B cells [4]. There is a complicated battery of biochemical reactions in cross-linking of IgE molecules and their high affinity receptors, including calcium (Ca^2+^) influx and mitogen-activated protein kinases (MAPKs) activation [5]. MAPKs contain extracellular signal-regulated kinase (ERK), p38 kinase, and c-jun N-terminal kinase (JNK), the phosphorylation of which are crucial in the differentiation, activation, proliferation, and degranulation between the immunocytes [6]. With the function of IgE–FcεRI interaction, RBL-2H3 cells have been generally used to study food allergy [7].

Sea buckthorn (*Hippophaë rhamnoides* L.), planted in Eurasia, is extensively used as a natural material for food and food additives because of its exceptionally nutritive value [8]. Flavonoids have been found to have being anti-cancer, anti-oxidant, and anti-microbial properties [9]. It has been verified that flavonoids can prevent or alleviate metabolic complications developed by food, such as diabetes, adiposity, and inflammatory diseases [10]. Numerous previous studies have demonstrated that sea buckthorn flavonoids (SBF) are momentous active ingredients, including rutin, quercitrin, quercetin, isorhamnetin, kaempferol, and their glycosides [11]. A current study has shown that sea buckthorn flavonoids downregulated cytokines including IL-6, IL-1β, and TNF-α to exert the inhibitory effects [12]. Total flavonoids of sea buckthorn exerted an anti-inflammation effect by regulating the Th1/Th2 balance to improve atopic dermatitis-like lesions [13]. Another study indicated that total flavonoids can ameliorate airway inflammation, and the main components including quercetin and isorhamnetin have potent binding affinities to MAPK kinase [14]. Karuppagounder et al. [15] summarized anti-inflammatory properties of quercetin in atopic dermatitis. It was shown that quercetin inhibited inflammatory cytokines and signaling pathway expression to play an anti-allergic role. Quercetin and kaempferol suppressed IL-4 and TNF-α release from RBL-2H3 cells and reduced development of degranulation, which can also maintain the intestinal barrier [16]. Isorhamnetin can achieve this as well; Wu et al. [17] found that isorhamnetin induced the release of histamine and IL-5, and restored and promoted the intestinal epithelial tight junction function in the allergic model based on ω-5 gliadin-derived peptide. However, the influence mechanism of anti-allergy with SBF, purified sea buckthorn flavonoid (PSBF), and their compounds are still unknown. Additionally, it is crucial to understand what kind of approach to take to retain flavonoids from sea buckthorn.

Some traditional processes for extracting flavonoids from plants usually need an extended period of time and affect the stability of the active composition, which can lead to the loss of flavonoids because of ionization, hydrolysis, and oxidation [18]. With water, hydrophilic, and lipophilic organic solution as solvent, high hydrostatic pressure (HHP) operates at pressures ranging from 100 to 600 MPa at a low temperature, which makes it very easy to extract thermolabile and unstable compounds [19,20]. Furthermore, as an easy and efficient method for extraction of flavonoids, HHP can significantly increase the extraction yield together with lower operation temperature, less time, and less consumption of solvent [18]. Up to now, there have been few studies on the employment of HHP for extracting flavonoids from sea buckthorn. 

This study explored the effect of HHP technology in SBF extraction and the influence and mechanism of anti-allergy mechanism. Thus, a cellular degranulation test was used to assess allergic reactions. In this work, the cellular degranulation and MAPKs pathway in RBL-2H3 cell lines were observed for the anti-allergic effect of SBF, PSBF, and their flavonoid compounds.

## 2. Materials and Methods

### 2.1. Reagents and Materials

Piperazine N, N0-bis (2-ethanesulfonic acid) PIPES, mouse anti-dinitrophenol (DNP) IgE, 4-nitrophenyl-N-acetyl-β-D-glucosaminide, and DNP bovine serum albumin (BSA) were purchased from Sigma-Aldrich Chemical Co. (St. Louis, MO, USA). Primary antibodies against the following proteins were used: phosphorylated p38 (Thr180/Tyr182), ERK1/2 (Thr202/Tyr204), and JNK (Thr183/Tyr185) as well as total p38, ERK1/2, JNK, and β-actin, which were purchased from Cell Signaling Technology Inc. (Beverly, MA, USA). The IgE, histamine, interleukin-4 (IL-4), and interferon (IFN)-*γ* ELISA kits were purchased from Jianglai Biotechnology Co., Ltd. (Shanghai, China). Rutin, quercitrin, quercetin, isorhamnetin, and kaempferol were purchased from Shanghai Yuanye Co., Ltd. (Shanghai, China) and were >98% pure, as determined by high-performance liquid chromatography–mass spectroscopy. All compounds were dissolved in dimethyl sulfoxide (DMSO) and final concentrations of DMSO were adjusted to 0.1% (*v*/*v*) in incubation buffer.

### 2.2. Optimization of High Hydrostatic Pressure Extraction

The HHP process was carried out using the high hydrostatic pressure system (HPP. L2-600, Baotou Kefa High Pressure Technology Co., Ltd., Baotou, China), which has a cylindrical pressure chamber with a capacity of 600 L. Distilled water was used as the pressure-transmitting medium. The rate of pressure build-up was about 130 MPa/min, with an immediate release of the pressure (<3 s). Fully ripe berries of sea buckthorn cultivars “Fructus” of *Hippophae* ssp. were homogenized and lyophilized for 48 h after removing seeds. Heat-sealed sterile homogeneous bags containing some lyophilized powder sample and ethanol solvent at different concentrations were placed in the pressure vessel.

Based on a single-factor experimental design, four factors and three levels Box–Behnken design (BBD) and response surface methodology (RSM) were carried out for the optimal combination of this study. The effects of four factors on the dependent variable flavonoid yield (y, mg/g) were investigated: (A) pressure, (B) pressure-holding time, (C) ethanol concentration, and (D) liquid-to-solid ratio. The selection and range of these factors were based on the results of single-factor experimental data. Twenty-nine experiments were required; the experimental design is shown in Table 1.

### 2.3. Purification and HPLC Analysis of Sea Buckthorn

The extraction of flavonoids using macroporous resins has the advantages of wide application, simple and safe operation, and renewable utilization [21]. After being lyophilized, the SBF content was 5.04 mg/g. For purification, the supernatant concentrations of total flavonoids were adjusted to 0.2 mg/mL, pH was adjusted to 5.0, and the sample was loaded on a macroporous resin column (AB-8). The column was eluted with absolute ethanol to collect purified total flavonoid.

The chemical composition of sea buckthorn flavonoids was analyzed by HPLC. All HPLC samples were filtered through 0.22 μm membrane filter before injection. The column temperature was 30 °C and the injection volume was 10 μL. Solvent A: acetonitrile and solvent B: 0.4% aqueous phosphoric acid solution were used as the mobile phase for the analysis. The procedure used to elute the mobile phase is shown in Appendix A. The detection wavelength was monitored at 370 nm at a flow rate of 1.0 mL/min. Rutin, quercetin, isorhamnetin, and kaempferol were identified by comparison with standard retention times.

### 2.4. Mediator and Cytokine Release Assays of RBL-2H3 Cells

The rat RBL-2H3 cell line (ATCC#CRL-2256™) was obtained from the Chinese Academy of Sciences. Cells were cultured in MEM supplemented with 15% FBS, 1% 100 U/mL penicillin, and 100 μg/mL streptomycin. Cell density was adjusted to 2 × 10^5^ cells/mL at 37 °C (5% CO_2_/95% air atmosphere) before harvesting. Cells were seeded in a 96-well plate, incubated overnight with anti-DNP IgE for passive sensitization and washed with modified PIPES buffer (25 mM PIPES, pH 7.2, 119 mM NaCl, 5 mM KCl, 0.4 mM MgCl_2_·6H_2_O, 1 mM CaCl_2_, 5.6 mM glucose, 40 mM NaOH, and 0.1% BSA) in triplicate. Cells treated with modified PIPES buffer were negative control. Cells were treated with SBF, PSBF, and five compounds at the set concentrations for 1 h at 37 °C. After incubation, cells were stimulated with 60 μL DNP-BSA (10 μg/mL) for 30 min at 37 °C. After the incubation period, cell supernatants were collected for β-hexosaminidase release measurement as previously described by Wang et al. [7]. Toluidine blue dye was used to stain the cells [22]. Histamine, IL-4, and IFN-γ release were measured using ELISA kits according to the manufacturer’s instructions.

### 2.5. Measurement and Observation of Intracellular Ca^2+^ Levels ([Ca^2+^]_i_)

[Ca^2+^]_i_ was measured and observed using the calcium reactive fluorescence probe Fluo-3AM method according to the published methods [8]. RBL-2H3 cells (2 × 10^5^ cells/well) were cultured in 96-well black plates. and were sensitized with 0.5 μg/mL of anti-DNP IgE for 12 h. After cells were washed with PBS, they were then treated with different concentrations of samples for 1 h. The cells were sensitized with 0.5 μg/mL anti-DNP IgE for 12 h. Cells were rinsed with PBS and then treated with different concentrations of samples for 1 h, then rinsed three times with Hanks’ balanced salt solution (HBSS) buffer and incubated with Fluo 3-AM (200 μM) for 40 min. Finally, it was stimulated with DNP-BSA (10 μg/mL). The fluorescence intensity was measured with the aid of a fluorometric imaging plate reader (excitation 488 nm, emission 525 nm). The fluorescent intensity was measured directly with a fluorescent microscope (Lecia DM18, Shanghai Leica Instrument Co., Ltd., Shanghai, China).

### 2.6. Western Bolt

Western blotting was performed as described previously with some modifications [23]. All proteins were isolated from RBL-2H3 cells in various groups, whose protein concentrations were determined using the BCA Protein Assay Kit. Proteins were then separated by 12% SDS-PAGE and transferred to PVDF membranes. After sealing the membrane with 5% bovine serum albumin for 2 h, the membrane was incubated with primary antibody at 4 °C overnight. The secondary antibody was incubated for 2 h at room temperature. In the end, signals were visualized with enhanced chemiluminescence reagent.

### 2.7. Statistical Analysis

Data analysis was performed with SPSS 25.0 software (SPSS Inc., Chicago, IL, USA). Results are presented as mean ± SD and differences between values were analyzed for statistical significance using ANOVA with Duncan’s test. The regression analysis and the optimization of the RSM were analyzed with the software Design Expert 8.0.5 (Stat-Ease Inc., Minneapolis, MN, USA). Adequacy of models was assessed by goodness-of-fit, coefficient of determination (R2), adjusted coefficient of determination (adj. R^2^), coefficient of variation (C.V.), and F-values. For all comparisons, ** p* < 0.05 and *** p* < 0.01 were used to determine statistical significance.

## 3. Results

### 3.1. Effect of HHP Condition on SBF Amount

As the non-thermal technology, HHP can protect the sensitive molecules with mild temperatures and pressure holding time [24]. The contents of flavonoids in sea buckthorn could be affected by the HHP conditions, such as pressure, pressure-holding time, ethanol concentration liquid-to-solid ratio, pressure, and dwell time. First of all, the influence of each individual factor was explored through single-factor experiments and results of the ANOVA are shown in Figure 1. High pressures from 200 to 600 MPa were applied with pressure-holding time of 8 min, ethanol concentration of 70%, and a constant ratio of liquid-to-solid of 30 mL/g. As seen in Figure 1A, when the pressure increased from 200 MPa to 400 MPa, the initial value gradually increased, and the amount reached the highest value of 4.15 mg/g at 400 MPa. When the pressure was over 400 MPa, the SBF amount was decreased. Therefore, the pressure of 400 MPa was deemed to be the optimum extraction pressure standard. The effect of pressure-holding time on extraction amount of SBF is exhibited in Figure 1B. Firstly, it was set at 4, 6, 8, 10, and 12 min of pressure-holding time while other extraction parameters were given as follows: pressure at 400 MPa, ethanol concentration of 70%, and liquid-to-solid ratio 30 mL/g. During the extraction time from 4 to 10 min, the amounts were increasing; after that, it decreased 10 min later. It could be seen that the extraction process of active ingredients by high hydrostatic pressure method was fast, and the active ingredients could quickly diffuse into the solvent to achieve equilibrium. Therefore, the pressure-holding time of 10 min was selected for the further study.

As described in Figure 1C, the effect of ethanol concentration on extraction amount of SBF was straightforward. With the increase in ethanol concentration from 50% to 70%, the SBF extraction amount increased. When ethanol concentration reached 70%, the maximum extraction amount was 4.82 mg/g. Therefore, ethanol concentration of 70% was selected for subsequent study. The effect of liquid-to-solid ratio on SBF extraction amount is shown in Figure 1D. With an increase in liquid-to-solid ratio from 10 to 40 mL/g, the amount of SBF firstly increased and then decreased. The liquid-to-solid ratio of 40 mL/g was sufficient to reach the high extraction amount.

### 3.2. Optimization of the HHP Condition and Validation of the Model

Appendix A presents the experimental conditions and results for 29 runs. The F-value of this model was 31.21, which implies the model has a highly significant difference (*p* < 0.001). Moreover, the A, C, A^2^, C^2^, and D^2^ were significant points. The model short of fit was not significant (*p* > 0.05), which made clear that the relationship between the independent variables and dependent responses were explained in the developed model [18]. The model correlation coefficient was R^2^ = 0.9690, indicating that the agreement between the experimental values and the theoretical model was 96.90%.

In Figure 2A, when the ethanol concentration was 70% and liquid-to-solid ratio were kept at 30 mL/g, the SBF amount accrued along with increasing pressure and pressure-holding time. The interaction effects of any other extraction variables on the SBF were similar (Figure 2B–F). By applying a multiple regression analysis, the relationship between the tested independent variables and the response is explained in Equation (1).
Y = 4.98 + 0.327 A + 0.0746 B + 0.02825 C + 0.0146 D − 1.18 A^2^ − 0.2326 B^2^ − 0.6451 C^2^ − 0.4427 D^2^(1)

The optimal conditions for SBF extraction were as follows: pressure of 415.25 MPa, pressure-holding time of 10.26 min, ethanol concentration of 72.18%, and liquid-to-solid ratio of 39.66 mL/g. Considering the actual operation conditions, the optimal extraction conditions were revised, that is, extraction pressure of 415 MPa, pressure holding time of 10 min, ethanol concentration of 72%, and the liquid–material ratio of 40 mL/g. To examine the validity of the model, an extraction was performed with three replicates under these conditions for examining the validity of the model. The measured values (5.05 mg/g) lay within a 95% mean confidence interval of the predicted value (5.04 mg/g), which confirmed the predictability of the model.

### 3.3. Identification and Quantified of SBF and PSBF

The SBF was extracted with HHP technology. The PSBF was a compound obtained by purification of SBF with an AB-8 macroporous resin column. SBF and PSBF contain a large amount of flavonol glycosides [25], such as rutin, quercitrin, quercetin, isorhamnetin, and kaempferol. We analyzed SBF and PSBF extracted from sea buckthorn by column chromatography; the results are shown in Figure 3 and Table 2. It could be seen that quercitrin in SBF and PSBF was 945.28 and 4105.66 μg/mL, respectively, which was the predominant flavonoid in the berries of sea buckthorn. In addition, the concentrations of rutin, quercetin, isorhamnetin, and kaempferol in SBF were 123.72, 23.23, 15.15, and 0.82 μg/mL, respectively.

### 3.4. SBF, PSBF, and Its Active Compounds Inhibit Degranulation in RBL-2H3 Cells

To investigate the influence of SBF, PSBF, and five compounds on the activity of RBL-2H3 cells, MTT assay was used to test the cytotoxicity. With diverse concentrations of SBF, PSBF, and five compounds dosing cells for 4 h, no cytotoxicity was observed (Appendix A). Thus, the experiment results indicate that SBF, PSBF at concentrations of 25–100 μg/mL, and the five compounds at concentrations of 0–40 μmol/L are applicable in vitro studies.

Flavonoids play an anti-allergic, anti-inflammatory, and anti-oxidant role by promoting wound healing and protecting the gut [26]. Firstly, the effects of SBF, PSBF, and five compounds on degranulation was assessed in RBL-2H3 cells sensitized by IgE. β-hexosaminidase and histamine, as main allergic mediators, were released by granulated mast cells stimulated with IgE [23]. Quercitrin at 40 μmol/L suppressed β-hexosaminidase release to a greater degree than quercetin, isorhamnetin, and kaempferol (Figure 4A–E). Antigen-stimulated RBL-2H3 cells treated with SBF significantly reduced β-hexosaminidase release and the inhibition ratio has a dose-dependent effect; in contrast, PSBF showed a converse trend (Figure 4F). The release of histamine was measured to further explore the inhibitory effects of SBF, PSBF, and five compounds on mast cell. SBF, PSBF, and five compounds turned out to markedly decrease histamine secretion in antigen-stimulated RBL-2H3 cells (Figure 4G–L), especially quercitrin, quercetin, and isorhamnetin at 40 μmol/L, SBF at 100 μg/mL, and PSBF at 25 μg/mL. Cruz et al. [27] found that extract of *Kalanchoe pinnata* could relieve degranulation and histamine secretion by antigen-induced mast cells, which proves that quercitrin is a critical component of extracts against the allergic reaction.

Following allergen stimulation and SBF, PSBF, and five compounds administration, the morphology of RBL-2H3 cells were changed after degranulation. As a synthetic basic dye, Toluidine blue can integrate with substances including acidic materials, histamine, and other heterochromous components secreted by mast cells, and then different colors can be observed [22]. After staining with toluidine blue dye in RBL-2H3 cells, the morphological changes were observed under an optical microscope. As shown in Figure 5A, the cells developed and refraction occurred normally in the control group, which manifested as long shuttle forms, and the numerous intracytoplasmic granule content was uniformly distributed in appearance. In contrast, there were typical degranulation morphology of the cells in allergy group, whose specific performance included irregular form, and remote and disorderly granule content within the cell cytoplasm (Figure 5B). All performance indicated that RBL-2H3 cells were undergoing the process of degranulation. RBL-2H3 cells treated with 40 μmol/L quercitrin, 100 μg/mL SBF, 25 μg/mL PSBF, and 100 μmol/L ketotifen fumarate were found to restore and elongate in shape, and increase the intracytoplasmic granule content (Figure 5C–F), which proved treatment inhibited degranulation in cells. In addition, treatments of quercitrin and SBF revealed a superior suppression on degranulation (Figure 5D,E).

### 3.5. SBF, PSBF, and Its Active Compounds Regulate Release of IL-4 and IFN-γ

IL-4 and IFN-*γ* are inflammatory cytokines and make a contribution in inflammatory and allergic disorders [28]. Thus, we investigated the effects of SBF, PSBF, and five compounds on IL-4 and IFN-γ release in RBL-2H3 cells sensitized by IgE. As observed in Figure 6, after being sensitized with DNP-specific IgE antibody and provoked by antigen DNP-BSA, RBL-2H3 cells secreted plenty of IL-4, which increased significantly (*p* < 0.01) while IFN-γ production reduced significantly (*p* < 0.05). Meanwhile, 40 μmol/L quercitrin, isorhamnetin, and 100 μg/mL SBF significantly reduced cellular IL-4 levels, while 40 μmol/L quercitrin and 100 μg/mL SBF were the only compounds that could observably (*p* < 0.01) increase the levels of IFN-*γ* (Figure 6). The proportion of IFN-*γ* to IL-4 levels secreted by cells was generally employed as the standard to determine the polarization status of Th1/Th2 cells. The results reveal that the ratio of IFN-γ/IL-4 of RBL-2H3 cells in the allergy model group was 46.70, while the ratio in the control group was 61.78 (Figure 6M). When the cells were treated with 40 μmol/L quercitrin and 100 μg/mL SBF, the ratio of IFN-γ/IL-4 was 62.25 and 64.31, respectively. However, there were no significant differences between that in the positive group and the control group (Figure 6M). It indicates that RBL-2H3 cells with sensitization and stimulation treated with 40 μmol/L quercitrin and 100 μg/mL SBF could recover the ratio of IFN-*γ*/IL-4, which resulted in the polarization of Th1/Th2 cells towards normality and the Th1/Th2 balance being regulated.

### 3.6. SBF, PSBF, and Its Active Compounds Inhibit Intracellular [Ca^2+^]_i_ Influx

It is important for mast cells to test the degranulation process of calcium ions. As a fluorescent probe, the fluo-3AM served as a carrier of calcium signal, who can be loaded into RBL-2H3 cells [29]. Therefore, the effects of SBF, PSBF, and its active compounds on extracellular Ca^2+^ influx in RBL-2H3 cells treated with sensitization and stimulation were studied through fluo-3 AM. As shown in Figure 7A−G, when RBL-2H3 cells were stimulated by antigen DNP-BSA, the levels of [Ca^2+^]_i_ increased sharply. Compared to the allergy model group, [Ca^2+^]_i_ of cells treated with SBF, PSBF, and its active compounds were always lower within 0–400 s. In addition, [Ca^2+^]_i_ in 40 μmol/L quercetin and 100 μg/mL SBF treated groups were similar to that in the control group. As exhibited in Figure 7H–M, the average value of [Ca^2+^]_i_ in RBL-2H3 cells appeared within 400 s, which indicates that the average value of [Ca^2+^]_i_ in the cells with sensitization and stimulation markedly climbed (*p* < 0.01). SBF, PSBF, and its active com-pounds lessened the enhancement in [Ca^2+^]_i_ caused by antigen stimulation. In particular, 40 μmol/L quercetin, and 100 μg/mL SBF and PSBF played the best role in inhibiting the augment of [Ca^2+^]_i_.

The RBL-2H3 cells with sensitization and stimulation and then treatment with 40 μmol/L quercetin, and 100 μg/mL SBF and PSBF were observed with the inverted fluorescence microscope investigation. As shown in Figure 8, cells with sensitization and stimulation show a sharper increase in [Ca^2+^]_i_ levels than the control group; simultaneously, when treated with 100 μg/mL SBF and 40 μmol/L quercetin, the level of [Ca^2+^]_i_ decreased nearly to the level of the positive group. This result indicates that SBF might restrain the extracellular [Ca^2+^]_i_ influx to inhibit the degranulation of RBL-2H3 cells.

### 3.7. SBF, PSBF, and Its Active Compounds Inhibit the MAPK Signaling Pathway

The surface receptor FcεRI in mast cells can bind to the IgE that is cross-linked with the antigen and then it is activated, which mediates degranulation and induces MAPK activation. Therefore, the effect of SBF, PSBF, and active compounds and the roles they play in the MAPK pathway in mast cells were explored. To determine the downstream signaling molecules, we examined the phosphorylation of ERK, JNK, and p38, for that affects the degranulation and production of proinflammatory mediators. As shown in Figure 9, SBF, PSBF, and the compounds significantly reduce the phosphorylated level of p38 protein in the MAPK signaling cascade. SBF, PSBF, and these compounds (especially quercitrin) also inhibit phosphorylation of MAPK activation in RBL-2H3 cells with sensitization and stimulation. However, among all compounds, only 100 μg/mL PSBF downregulated (*p* < 0.01) phosphorylation of ERK. Furthermore, 100 μg/mL SBF, PSBF, and 40 μmol/L rutin, quercitrin, quercetin, and isorhamnetin markedly reduced the phosphorylation of p38 and JNK.

## 4. Discussion

Figure 4 shows the interaction among the pressure, pressure-holding time, ethanol concentration, and the liquid-to-solid ratio. HHP with high-pressures and short-time operation can cause deformation and destruction of the plant cell structures, which will significantly affect the diffusion of the bioactive compounds and accelerate the dissolution [18]. Up to now, many researchers have found that increasing the extraction pressure could improve the penetration rate of solvent into the solid tissue and greatly shorten the extraction time [30]. However, excessive pressure could destroy the structure of flavonoids, lowering the extraction amount [31]. In line with previous studies, our research shows that the initial value gradually increased during the pressure increased from 200 MPa to 400 MPa, and the SBF amount was decreased when the pressure was over 400 MPa, while the maximum amount was 4.15 mg/g that was reached at 400 MPa exactly. The high hydrostatics pressure extraction method has the advantages of being fast and efficient, and is a process of uniform and stable treatment pressure and instant transfer [32]. Our results were broadly similar in that the amounts increased when the extraction time increased from 4 to 10 min and it decreased 10 min later. Therefore, the pressure-holding time of 10 min was selected for the further study. When ethanol concentration was greater than 70%, the extraction amount decreased, which might be because some alcohol-soluble substances and pigments dissolve out with the increase in ethanol concentration and compete with flavonoids [33]. The amount of SBF firstly increased with an increase in liquid-to-solid ratio from 10 to 40 mL/g, and then the extraction decreased when the liquid-to-solid ratio was over 40 mL/g. The liquid-to-solid ratio of 40 mL/g was sufficient to reach the high extraction amount. A similar pattern of results was obtained in the extraction of pectin from navel orange peel through HHP by Sang et al. [30]. Therefore, we adopted extraction at 415 MPa for 10 min, 72% ethanol concentration, and a liquid-to-solid ratio of 40 mL/g for further study objects.

Some studies have focused on the extraction of flavonoid and its compounds. Gabriele et al. [34] used hot water to extract flavonoids from sea buckthorn leaves, and the amount they achieved was 2.98 µg/mL. Compared to HHP extraction, the production by hot water extracting method was much less. Moreover, when the temperature is too high, the flavonoid is not stable and may be broken down or change the activity. In addition, the contents of flavonoids in fresh sea buckthorn berries from various areas were analyzed, which ranged from 39.5 to 240.2 mg/100 g [35]. It was also particularly less than flavonoid in this study. Therefore, HHP extraction improved the flavonoids’ amount and protected its activity as much as possible. As beneficial natural active ingredients in nature, flavonoids are distributed in many plants, which spread in Selaginella sinensis. Microwave-assisted extraction-IL was used for extracting amentoflavone and hinokiflavone. The results show that the extraction yields of two kinds of flavonoids were 1.96 mg/g and 0.79 mg/g, respectively [36]. On the basis of previous studies, it can be found that HHP extraction of flavonoids has great advantages and development prospects.

Nowadays, researchers have focused on alleviating food allergy through various natural plant extracts. This study illustrated the changes in the allergenic properties of IgE-sensitized RBL-2H3 cells. The widely used cell model for detecting type I hypersensitivity is IgE-mediated RBL-2H3 cells [37]. The degranulation components including allergenic mediators and cytokines secreted by cells are important criteria in expressing the food allergy response [38]. Furthermore, as markers of the degranulation in mast cells, β-hexosaminidase and histamine play a crucial part in allergic diseases [23]. Our experiments confirm that SBF, PSBF, and their active compounds do indeed suppress the degranulation of activated mast cells via reducing the release of β-hexosaminidase and histamine, which is consistent with previous results of Itoh et al. [39] and Beáta et al. [40], who indicate that kaempferol, quercitrin, quercetin, other flavonoids, and natural plant extracts, as well as abundant plant components in fruits and vegetables, likely have anti-allergic properties.

For instance, quercetin, quercitrin, and kaempferol can inhibit degranulation of RBL-2H3 cells by reducing the release of β-hexosaminidase, histamine, and IL-4. Accordingly, IgE-mediated allergic reactions were suppressed [5,41]. Hence, we conjecture that these compounds in SBF may have an alleviative function on allergic reactions. From the result, it is clear that 40 μmol/L quercitrin, isorhamnetin, and 100 μg/mL SBF significantly reduced cellular IL-4 levels, while only 40 μmol/L quercitrin and 100 μg/mL SBF could significantly (*p* < 0.01) increase IFN-*γ* levels (Figure 6). IFN-*γ,* which can activate antigen-presenting cells, is an essential anti-inflammatory and can regulate Th1 differentiation [8]. IL-4, as a pro-inflammatory factor and pleiotropic cytokine necessary for Th2 cell response, is of vital importance in differentiation of primitive naive T cells into Th2 cells and allergic reactions [42]. The ratio of IFN-*γ*/IL-4 revealed that the cells treated 40 μmol/L quercitrin and 100 μg/mL SBF with the positive control group made no difference. This implies that it is significant to keep the balance between Th1 cytokines and Th2 cytokines in allergic disorders [43].

The second messenger of intracellular signal transduction including Ca^2+^ can play an important role in mast cells degranulation [8]. Along with the mast cells degranulation and inflammatory mediators’ production, Ca^2+^ influx is increased in cells [44]. In this study, it is found that cells treated with 40 μmol/L quercetin, 100 μg/mL SBF and PSBF markedly inhibit the influx of extracellular Ca^2+^. Meanwhile, compared to the model group, the intracellular fluorescence intensity of these groups significantly decreased. These findings suggest that because of inhibiting extracellular Ca^2+^ influx in RBL-2H3 cells with sensitization and stimulation, quercetin, SBF, and PSBF, at some concentrations, influence degranulation via the Ca^2+^-dependent pathway.

Moreover, the effect of SBF, PSBF, and the compounds on the MAPK pathway in IgE-sensitized mast cells was studies (such as IFN-*γ*) [37,45]. Firstly, Lyn activation was mediated by Lyn phosphorylation of tyrosine residues, which were able to trigger a cascade of signaling events and then recruited and activated other kinases including Syk [46]. The activation of Syk contributed to an increase in intracellular Ca^2+^ and the activation of MAPK. SBF, PSBF, and the compounds (especially quercitrin) also inhibited phosphorylation of MAPK activation in antigen-stimulated RBL-2H3 cells. The current study certificates that quercitrin alleviates the colitis model by enhancing tumor necrosis factor α (TNF-α) and interleukin 1β (IL-1β) expressions [47]. In ovariectomized mice, quercitrin decreased the expression of p-INK, p-ERK, and p-p38 to blunt the MAPK signaling pathways [48]. Similarly, the crucial flavonoid from lotus leaf can mediate brown-adipocyte differentiation and thermogenic via inhibiting the p38 MAPK signal pathway [49]. Chung et al. [50] found that quercetin could markedly inhibit the IgE–antigen complex that stimulates phosphorylation of the p38, ERK, and JNK proteins. In addition, quercitrin extracted from *Houttuynia cordata* Thunb reduced inflammation both in vitro and in vivo models [51]. This is in good agreement that SBF and various compounds inhibited the production of inflammatory mediators in antigen-stimulated mast cells by weakening MAPK activation. Consequently, all these results reveal that sea buckthorn flavonoids have potent antiallergic activity.

## 5. Conclusions

Overall, we obtained the optimal conditions for the highest SBF amount (5.05 mg/g) of sea buckthorn. The rutin, quercitrin, quercetin, isorhamnetin, and kaempferol in SBF and PSBF were identified and quantified by HPLC analysis. In addition, the RBL-2H3 cell models reveal that SBF, PSBF, and the five compounds suppress degranulation, and the generation of IL-4, as well as the extracellular Ca^2+^ influx. The 40 μmol/L quercitrin, kaempferol and 100 μg/mL SBF could regulate Th1/Th2 cytokine balance; 40 μmol/L quercitrin and 100 μg/mL PSBF reduced the extracellular Ca^2+^ influx. In addition, 100 μg/mL SBF, PSBF, and 40 μmol/L rutin, quercitrin, quercetin, and isorhamnetin markedly reduced the phosphorylation of p38 and JNK, thereby suppressing the phosphorylation of the MAPK signaling pathways. Based on these results, HHP is a promising technology for extracting flavonoids and SBF might be a promising anti-allergic therapeutic agent.

## Figures and Tables

**Figure 1 foods-13-00560-f001:**
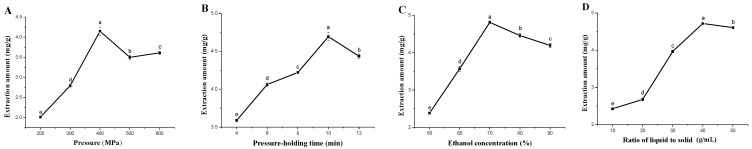
Effect of pressure (**A**), pressure−holding time (**B**), ethanol concentration (**C**), ratio of liquid−to−solid (**D**) on high hydrostatics pressure extraction (HHP) amount of flavonoid from sea buckthorn. Different letters represent significant differences (*p* < 0.05).

**Figure 2 foods-13-00560-f002:**
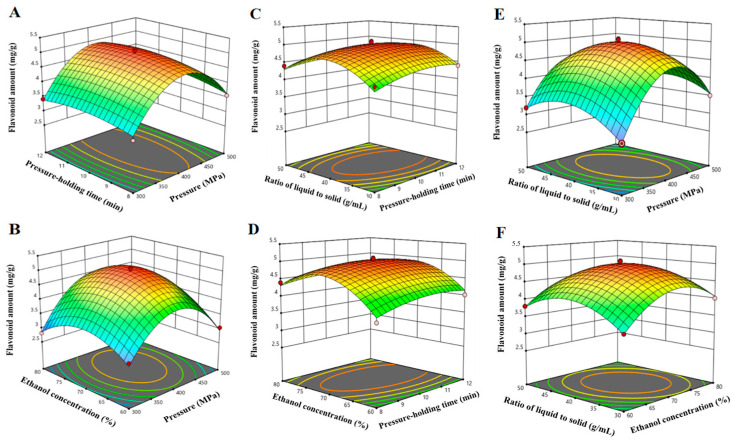
The plots for 3D response surface reveal the effect of (**A**) pressure and pressure-holding time; (**B**) pressure and ethanol concentration; (**C**) pressure-holding time and ratio of liquid-to-solid; (**D**) pressure-holding time and ethanol concentration; (**E**) pressure and ratio of liquid-to-solid; (**F**) ethanol concentration and ratio of liquid-to-solid on the extraction amount of flavonoid.

**Figure 3 foods-13-00560-f003:**
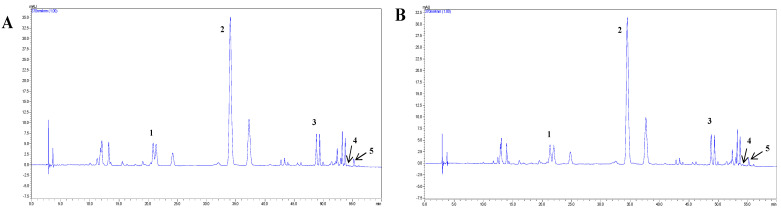
The chemical structures of flavonoids components and HPLC chromatogram of SBF (**A**) and PSBF (**B**) obtained by HHP under optimum conditions (415 MPa, ethanol concentration 72%, liquid-to-solid ratio 40 mL/g, time 10 min, and temperature 20 °C). Peaks 1, 2, 3, 4, and 5 represent rutin, quercitrin, quercetin, kaempferol, and isorhamnetin, respectively.

**Figure 4 foods-13-00560-f004:**
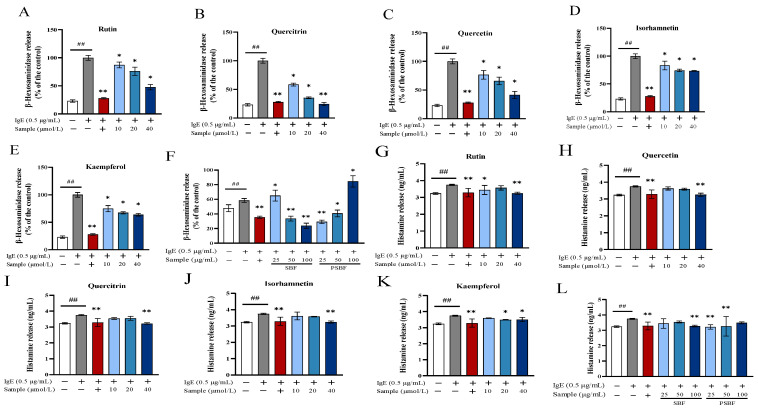
Effect of SBF, PSBF, and five compounds on β-hexosaminidase and histamine release in RBL-2H3 cells. The positive control is ketotifen fumarate at 100 μmol/L. The β-hexosaminidase (**A**–**E**) and histamine (**G**–**K**) release in RBL-2H3 cells treated with 10, 20, and 40 μmol/L of rutin, quercitrin, quercetin, isorhamnetin, and kaempferol. The β-hexosaminidase (**F**) and histamine (**L**) release in RBL-2H3 cells treated with 25, 50, and 100 μg/mL of SBF and PSBF. Results are expressed as the mean ± SD of three independent experiments. * *p* < 0.05 and ** *p* < 0.01, in comparison with allergy model group; ## *p* < 0.01 in comparison with the control group.

**Figure 5 foods-13-00560-f005:**
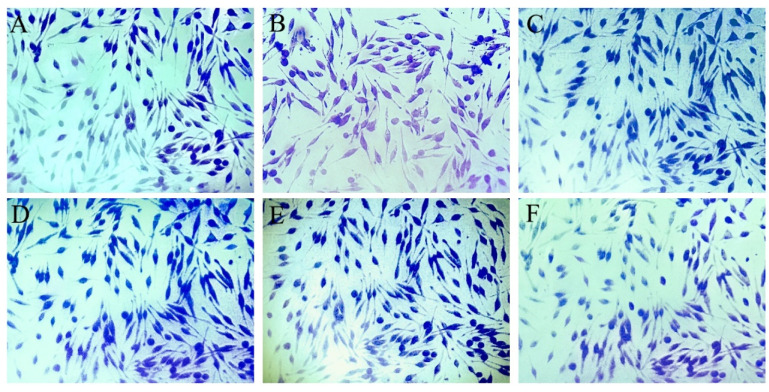
Effects of SBF, PSBF, and five compounds on the morphology of RBL-2H3 cells degranulation under an inverted microscope. (**A**) Control; (**B**) allergy model; (**C**) positive control; (**D**) 40 μmol/L of quercitrin; (**E**) 100 μg/mL of SBF; (**F**) 25 μg/mL of PSBF.

**Figure 6 foods-13-00560-f006:**
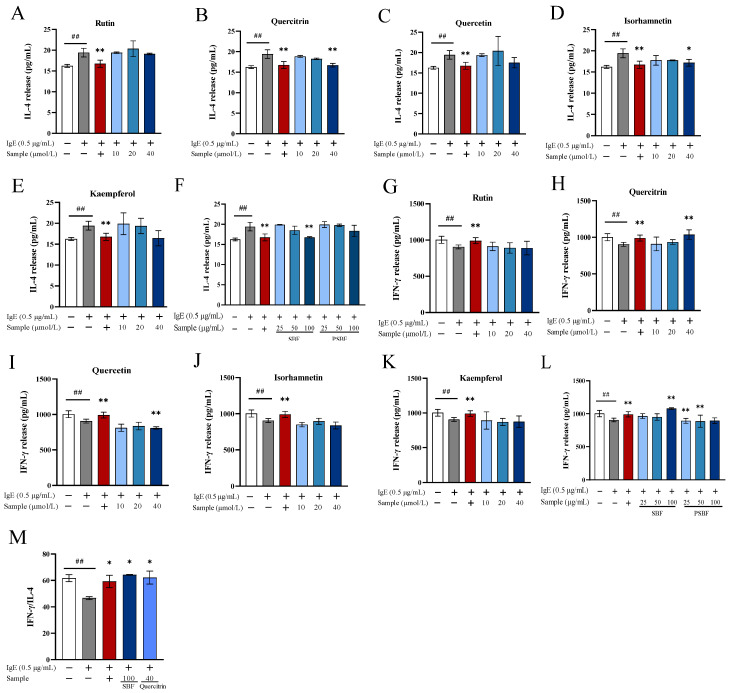
Effect of SBF, PSBF, and five compounds on IL-4, IFN-*γ* release, and Th1/Th2 balance in RBL-2H3 cells. The positive control is ketotifen fumarate at 100 μmol/L. The IL-4 (**A**–**E**) and IFN-*γ* (**G**–**K**) release in RBL-2H3 cells treated with 10, 20, and 40 μmol/L of rutin, quercitrin, quercetin, isorhamnetin, and kaempferol. The IL-4 (**F**) and histamine (**L**) IFN-*γ* in RBL-2H3 cells treated with 25, 50, and 100 μg/mL of SBF and PSBF. The ratio of IL-4 and IFN-*γ* (**M**) in RBL-2H3 cells treated with 40 μmol/L of quercitrin and 100 μg/mL of SBF. Results are expressed as the mean ± SD of three independent experiments. (## *p* < 0.01, * *p* < 0.05, and ** *p* < 0.01).

**Figure 7 foods-13-00560-f007:**
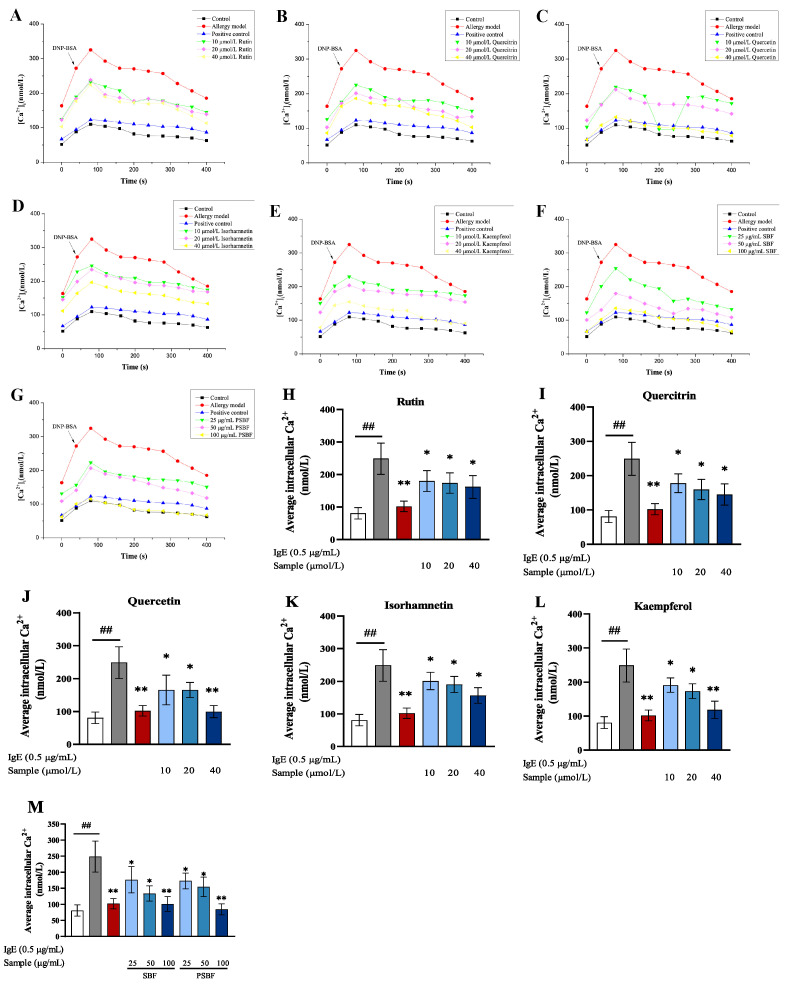
Effect of SBF, PSBF cellular Ca^2+^ in RBL-2H3 cell. (**A**) Rutin; (**B**) quercitrin; (**C**) quercetin; (**D**) isorhamnetin; (**E**) kaempferol; (**F**) SBF; (**G**) PSBF. Effect of five compounds (**H**–**L**) and SBF, PSBF (**M**) on average intracellular Ca^2+^ in RBL-2H3 cells. The positive control is ketotifen fumarate at 100 μmol/L. Results are expressed as the mean ± SD of three independent experiments. (## *p* < 0.01, * *p* < 0.05 and ** *p* < 0.01).

**Figure 8 foods-13-00560-f008:**
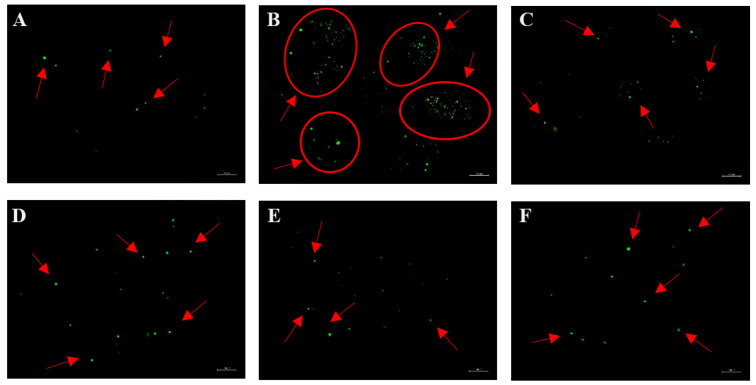
Effect of SBF, PSBF, and quercetin on intracellular Ca^2+^ in RBL-2H3 cells was observed on an inverted fluorescence microscope. The positive control is ketotifen fumarate at 100 μmol/L. (**A**) Control; (**B**) allergy model; (**C**) positive control; (**D**) 40 μmol/L of quercetin; (**E**) 100 μg/mL of SBF; (**F**) 100 μg/mL of PSBF.

**Figure 9 foods-13-00560-f009:**
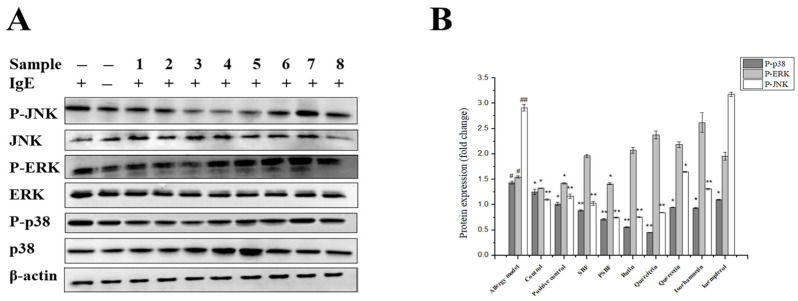
Effect of SBF, PSBF, and five compounds on MAPK signaling in RBL-2H3 cells. The positive control is ketotifen fumarate. (**A**) p38, ERK, JNK, and phosphorylation of p38, ERK, JNK were determined with Western blot. Number 1 represents ketotifen fumarate at 100 μmol/L, and 2, 3 represent SBF and PSBF at 100 μg/mL. Number 4, 5, 6, 7, and 8 represent rutin, quercitrin, quercetin, isorhamnetin, and kaempferol at 40 μmol/L, respectively. (**B**) Densitometric analysis. Results are expressed as the mean ± SD of three independent experiments. * *p* < 0.05 and ** *p* < 0.01, in comparison with allergy model group; # *p* < 0.05, ## *p* < 0.01 in comparison with the control group.

**Table 1 foods-13-00560-t001:** Factors and levels of response surface test.

Code	Factor			
	A Pressure (MPa)	B Pressure-Holding Time (min)	C Ethanol Concentration (%)	D Liquid-to-Solid Ratio (mL/g)
−1	300	8	60	30
0	400	10	70	40
1	500	12	80	50

**Table 2 foods-13-00560-t002:** Comparison of the flavonoid contents (μg/mL) in SBF and PSBF.

Compound	Concentration (μg/mL)	
	SBF	PSBF
Rutin	123.72 ± 8.02	521.83 ± 18.73
Quercitrin	945.28 ± 20.32	4105.66 ± 192.15
Quercetin	23.23 ± 1.35	100.30 ± 8.34
Isorhamnetin	15.15 ± 0.87	75.10 ± 6.12
Kaempferol	0.82 ± 0.01	3.45 ± 0.61

## Data Availability

The original contributions presented in the study are included in the article, further inquiries can be directed to the corresponding author.

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
