# Peer review of "Sea Buckthorn Flavonoid Extracted by High Hydrostatic Pressure Inhibited IgE-Stimulated Mast Cell Activation through the Mitogen-Activated Protein Kinase Signaling Pathway"

_foods, 2024, doi:10.3390/foods13040560_

Round 1
Reviewer 1 Report
Comments and Suggestions for Authors
The paper entitled “Sea buckthorn flavonoid extracted by high hydrostatic pressure inhibited IgE-stimulated mast cell activation through the MAPK Signaling Pathway” is focused on the recovery of flavonoids bioactive compounds from a natural source as sea buckthorn. High hydrostatic pressure is used for the extraction and the main parameters affecting the process are optimized by Response Surface Methodology. The resulted extracts are used for studying their anti-allergy potential.
The aims of the study are clearly expressed.
The experimental program is described in such manner that it can be easily applied.
The obtained results are concise, clearly presented and discussed.
Conclusions are drawn according to the obtained data.
The authors will find bellow some corrections and adjustments that should be addressed.
- There are phrases rather difficult to understand (e.g. lines 16-18, 57-58, 111-112, 126-127, 146-147, 205-207 etc.). It is recommended to revise them.
- Exposing the effect of HPP condition on SBF amount separately is not necessary. Instead, it is recommended to accentuate the discussion on the optimization process that includes the effect of each parameter.
- It is recommended to include in the „Discussion” section a more detailed comparison with other extracts obtained by other extraction methods and from different other plant sources.
Where tests effectuated on the extracts to sustain other possible uses aside as anti-allergic therapeutic agent? With what results?
Comments on the Quality of English Language
Minor editing of English language required
Author Response
Dear reviewer, please check the attachment. Thanks.

Reviewer 2 Report
Comments and Suggestions for Authors
This is an interesting article that need a small revision. Only few points should be addressed:
1) The Authors should explain the difference between the SBF and the PSBF.
2) How big are the concentrations of flavonoids in the Figure 4A-E ? 100, 200 and 400 or 10, 20 and 40 mikrom ?
3) The Authors should discuss the difference in the influence of SBF and PSBF on the release of beta-hexosaminidase (Figure 4F), histamine (Figure 4L), IL-4 (Figure 6F) and IFN-gamma (Figure 6L).
Author Response

(The authors gave the same response as above.)

Reviewer 3 Report
Comments and Suggestions for Authors
- The paper titled “Sea buckthorn flavonoid extracted by high hydrostatic pressure inhibited IgE-stimulated mast cell activation through the MAPK Signaling Pathway” described the conditions for obtaining maximum flavonoid content of sea buckthorn flavonoid (SBF) along with HPLC analysis of extracted flavonoids as well as anti-allergy in RBL-2H3 cells by SBF, PSBF and their flavonoid compounds was evaluated. The manuscript has potential but need following changes before consideration.
- Title is good but do not use abbreviations in the title like “MAPK”
- Abstract is written good but lack the flow as first write study background followed by study methodology and then numerical results for all parameters and particularly add study stakeholders for which findings are useful at the end of abstract
- Introduction; “Globally, the prevalence of food allergies is on the rise” is open ended and need to be completed
- Introduction provides general overview and lack information and material regarding the role of Sea buckthorn flavonoids in food allergy control potential. Compounds responsible and control mechanism needed to be added in the study
- Flavonoids are present in several food raw material but what is the specific logoic for SBF that need to be justified and added in introduction part
- Material and method section is written good however “Western bolt” sections need spell check, elaboration along with reference
- Results are written in a good way however add latest references to support findings
- Do not add abbreviations in the headings and try to elaborate headings as much as possible
- Conclusion needs better elaboration along with addition of numerical results as well as lack conclusion for the food allergy that was major hypothesis of the study. Carefully revise the conclusion considering study objective
Comments on the Quality of English Language
- Grammatical mistakes observed at several places so there is need to go through the paper for language and grammatical mistakes check
Author Response

(The authors gave the same response as above.)
